# Biodegradable Electrospun Scaffolds as an Emerging Tool for Skin Wound Regeneration: A Comprehensive Review

**DOI:** 10.3390/ph16020325

**Published:** 2023-02-20

**Authors:** Deepika Sharma, Shriyansh Srivastava, Sachin Kumar, Pramod Kumar Sharma, Rym Hassani, Hamad Ghaleb Dailah, Asaad Khalid, Syam Mohan

**Affiliations:** 1Department of Pharmacy, School of Medical and Allied Sciences, Galgotias University, Greater Noida 203201, India; 2Department of Pharmacology, Delhi Pharmaceutical Sciences and Research University (DPSRU), Sector 3 Pushp Vihar, New Delhi 110017, India; 3Department of Mathematics, University College AlDarb, Jazan University, Jazan 45142, Saudi Arabia; 4Research and Scientific Studies Unit, College of Nursing, Jazan University, Jazan 45142, Saudi Arabia; 5Substance Abuse and Toxicology Research Centre, Jazan University, Jazan 45142, Saudi Arabia; 6Medicinal and Aromatic Plants and Traditional Medicine Research Institute, National Center for Research, Khartoum P.O. Box 2404, Sudan; 7School of Health Sciences, University of Petroleum and Energy Studies, Dehradun 248007, India; 8Center for Transdisciplinary Research, Department of Pharmacology, Saveetha Institute of Medical and Technical Science, Saveetha Dental College, Saveetha University, Chennai 600077, India

**Keywords:** nanofibers, polymers, scaffolds, wound healing, electrospinning, nanoscaffold, biodegradable polymer

## Abstract

Skin is designed to protect various tissues, and because it is the largest and first human bodily organ to sustain damage, it has an incredible ability to regenerate. On account of extreme injuries or extensive surface loss, the normal injury recuperating interaction might be inadequate or deficient, bringing about risky and disagreeable circumstances that request the utilization of fixed adjuvants and tissue substitutes. Due to their remarkable biocompatibility, biodegradability, and bioactive abilities, such as antibacterial, immunomodulatory, cell proliferative, and wound mending properties, biodegradable polymers, both synthetic and natural, are experiencing remarkable progress. Furthermore, the ability to convert these polymers into submicrometric filaments has further enhanced their potential (e.g., by means of electrospinning) to impersonate the stringy extracellular grid and permit neo-tissue creation, which is a basic component for delivering a mending milieu. Together with natural biomaterial, synthetic polymers are used to solve stability problems and make scaffolds that can dramatically improve wound healing. Biodegradable polymers, commonly referred to as biopolymers, are increasingly used in other industrial sectors to reduce the environmental impact of material and energy usage as they are fabricated using renewable biological sources. Electrospinning is one of the best ways to fabricate nanofibers and membranes that are very thin and one of the best ways to fabricate continuous nanomaterials with a wide range of biological, chemical, and physical properties. This review paper concludes with a summary of the electrospinning (applied electric field, needle-to-collector distance, and flow rate), solution (solvent, polymer concentration, viscosity, and solution conductivity), and environmental (humidity and temperature) factors that affect the production of nanofibers and the use of bio-based natural and synthetic electrospun scaffolds in wound healing.

## 1. Introduction

The skin of the human body is a versatile organ that serves as a natural barrier against environmental elements. The skin is composed of three layers: the epidermis, dermis, and hypodermis, all of which protect the body’s internal organs from external threats, whether physical, synthetic, or natural. As the skin plays an important role in the body, any significant damage, including major or severe wounds, must be managed quickly and effectively to prevent further harm [1]. Wound mending has become a significant worldwide general well-being concern for many years. Insufficient skin harm treatment can possibly be deadly. Subsequently, broad exploration in this space has been directed, with an attention on the advancement of successful, helpful methods, and the innovation of novel dressing materials that can advance wound healing [2]. Contingent upon the seriousness and intensity of the injury, a sufficient dressing with viable injury site security and quick skin recovery is clinically vital for speeding up injury recuperation [3]. Covering the injured area with a dressing material that prevents dehydration and contamination has been part of wound care for a long time [4]. Wound recuperating medicines have advanced throughout the past 10 years, from straightforward balms and bandages to cutting edge multifunctional wound dressings. Ideally, a wound dressing should: possess anti-inflammatory characteristics, absorb wound exudates, protect the wound from bacterial infection, shield from external stimuli, induce cell proliferation to expedite healing, and promote tissue regeneration. It should keep the injury site appropriately moist and protect the tissue from further harm. In order to fulfil its role of preventing wound compression, a dressing must possess elasticity in cases where compression is not possible, as well as being composed of a material that is soft, compatible with the body, hypoallergenic, and non-toxic [5]. The use of multipurpose wound dressings accelerate the rate of healing. The electrospinning system has, as of late, acquired fame for the creation of biopolymer-based nanofibrous structures. Electrospinning is a typical, minimal expense and tunable strategy for making super fine filaments with unique properties. Electrospinning innovation has been generally utilized in biomedical materials, for example, tissue designing frameworks, wound dressings, and drug delivery systems. Nanofibers that are spun electrically, with diameters on a nanoscale, are suggested to be great wound dressings and tissue replacements since they look like the extracellular matrix (ECM) [6]. An assortment of regular biopolymers (e.g., cellulose) were electrospun into nanofibers to reproduce the local tissue network for wound mending. Collagen, hyaluronic acid, gelatin, and chitosan were employed to their fullest potential simultaneously. Synthetic bio-based polymers polylactides (PLAs) and polyhydroxyalkanoates (PHAs) are frequently used for the electrospinning of wound dressings. By electrospinning natural and synthetic polymers (derived from biological sources) in coaxial, multi-nozzle, or blend configurations, wound dressing mechanical degradation and/or morphological properties can be tailored [7]. This review aims to highlight and describe the recent developments in the electrospinning of various synthetic and natural biopolymers, emphasizing their applications in wound healing. An overview of the four stages of wound healing and the electrospinning process and its effects on fiber morphology are provided. Additionally, a summary of newly developed multifunctional electrospun structures for wound healing, including cellulose, chitosan, PHAs, and PLA, PCL, PES, PS, PAA, and thermoresponsive polymers such as Poly(N-isopropylacrylamide) (pNIPAAm) and poly(N-vinlycaprolactam) (PVCL), are presented and discussed. The review’s final section discusses the potential uses of electrospun biopolymer fibers for treating wounds. To locate relevant material, we selected keywords such as nanofibers, polymers, scaffolds, wound healing, electrospinning, nanoscaffold, and biodegradable polymer. The search for these terms was conducted in several relevant databases, including Google Scholar, PubMed, and Scopus. The range of research materials examined encompassed research articles, review articles, book chapters, and conference abstracts published between 2017 and 2023, resulting in a total of 147 published materials, as shown in Figure 1. The literature without specific and detailed data pertaining to the topic was excluded from this review.

## 2. An Overview of Wounds and Their Consequences

Various sources can lead to wounds, including surgery, trauma, external pressures, abrasions, and illnesses such as diabetes and cardiovascular diseases. Depending on the cause and the severity of the injury, these wounds can be classified as either acute or chronic [8]. An organized and appropriate repair process is typically followed by an acute wound, leading to the long-term restoration of anatomical and functional integrity. Chronic wounds, on the other hand, are unable to reach ideal anatomical and functional integrity. The healing process may be altered based on the characteristics of the patient and the environment, such as the type, intensity, and state of the injury, in addition to potential health conditions, such as vascular, metabolic, and autoimmune diseases, and the use of medications [9]. A wound that has been healed minimally is identified by the re-establishment of its anatomical makeup, yet without ongoing functional results; as a result, it can reappear. A perfectly healed wound is when the region returns to its original anatomy, performance, and outward appearance after an injury. A wound that has been fully healed is set apart from these two states by the formation of lasting anatomical and functional continuity. Numerous additional elements, including oxygen, infection, swelling, inflammation, trauma, and the body’s own system, can be used to evaluate a wound’s degree of healing. All of these characteristics can reveal information about the cause, pathophysiology, and state of a wound [10].

In conclusion, it is essential to think about how injuries shape the host, as patient health is essential in deciding how systemic components shape the wound. Since the mending procedure is ever-changing and requests a consistent, organized, and steady assessment, including continuous re-evaluation of wound range, type, and seriousness, it can be hard to gauge the recuperating process. Persistent wounds reduce the quality of life, and the cost of care is seen in morbidity and even mortality, as well as the psychological cost, and the lengthened hospital stay. These factors have led to wounds being referred to as a “Silent Epidemic” [11]. Most financial costs are a result of employing medical personnel; hospital stays, both in terms of length and expense; as well as the selection of materials and treatments. For all of these reasons, creating new technologies aimed at enhancing the healing process is challenging [12].

### 2.1. Chronic and Acute Wound Healing

The normal wound-healing process for acute injuries (such as those caused by trauma or surgery) creates an orderly and predictable pattern of tissue repair [13]. In comparison, the management of chronic wounds is chaotic, and they can primarily be classified as vascular ulcers (encompassing venous and arterial ulcers), diabetic ulcers, and pressure ulcers [14]. Chronic wounds that become inflamed for long periods of time lead to the growth of biofilm, the gathering of microorganisms, as well as the release of platelet-derived materials such as TGF-β and molecules from the extracellular matrix. The production of inflammatory cytokines, such as TNF-α and IL-1, over a prolonged period of time, causes a high amount of protease to be present in the wounded area. This amount surpasses the number of inhibitors, resulting in the breakdown of the extracellular matrix and fueling the inflammatory and proliferative phases of healing [15]. Chronic injuries also include phenotypic flaws in the cells and The accumulation of inflammatory cells in the wound bed of a chronic wound leads to an increase in the concentration of reactive oxygen species (ROS), causing damage to Extracellular Matrix (ECM) proteins and leading to the premature aging of the cells [16]. Chronic injuries also include phenotypic flaws in the cells and dermis, such as decreased growth factor receptor density and mitogen potential, which prevent local cells from properly responding to signals that promote wound healing [17].

### 2.2. Progression of Healing a Wound

The human body’s proceeding of wound healing is made up of four phases that are tightly linked and overlap with one another, the steps are: hemostasis, inflammation, proliferation, and remodeling [10].

#### 2.2.1. Hemostasis

Platelet adhesion and blood coagulation are the primary mechanisms for attaining hemostasis [18]. Clotting begins with tissue factors, which are typically away from the circulatory system. When damage happens to the vascular system, the tissue factors connect with factor VII, allowing the tissue factor-producing cells, such as adventitial fibroblasts, to be included in the blood clotting process. In turn, factors IX and X are activated by the TF: factor VII complex. The extrinsic pathway is the name provided to this process that activates factor X since it takes place outside the endothelium [19]. A minimal quantity of prothrombin is changed into thrombin when coagulation factor X (Xa) binds with its cofactor (factor Va). Platelet activation, a principal purpose of the initially transformed thrombin, prepares the path for the intrinsic route. The intrinsic pathway also activates factor X and is a concurrent and supportive mechanism. Once detached from factor VIII, the von Willebrand factor (vWf) helps further bind platelets to the wounded area [19]. Once the activated platelets connect with factors V, VIII, and IX, factor Xa is generated and added to the prothrombinase on the platelet’s surface, comprising of factors Xa and Va, which leads to a high level of thrombin and hastens the formation of the fibrin clot. The intrinsic and extrinsic pathways then come together, causing the production of fibrin subunits that link up, forming strands that bind the platelets and secure the plug in position [20]. The protein C/protein S/thrombomodulin (TM) system on endothelial cells is capable of deactivating factors Va and VIIIa, thus limiting clot formation to the damaged region. When platelets aggregate and degranulate, they release a range of substances, including platelet-derived growth factor (PDGF), transforming growth factor-β (TGFβ), transforming growth factor-α (TGFα), basic fibroblast growth factor (FGF), and insulin-like growth factor-I (IGF-I), which activate the inflammatory process. This is performed by employing chemotaxis, allowing monocytes and neutrophils to move from the blood vessels to the injury site [21]. Figure 2 depicts the various stages of wound healing.

#### 2.2.2. Inflammation

Following clot formation and blood loss regulation, different chemical mediators control the inflammatory phase. Cyclooxygenase 2 (COX-2) activates endothelial cells to produce prostaglandins, resulting in vasodilation, the breaking apart of platelets, and the formation of leukotrienes [22]. The process of exuding and cleaning up begins when the grouped platelets burst open and send out powerful chemo-attractants that draw in neutrophils, macrophages, and lymphocytes [10]. Neutrophils, which are drawn to the fibrin matrix by PDGF and other cytokines, are frequently the first cells to migrate. Bacteria, foreign matter, and cell debris are all phagocytosed by neutrophils. Neutrophils utilize proteolytic enzymes and reactive free oxygen radicals to clear away bacteria and any other detritus at the injury site. Furthermore, they discharge interleukin-1, interleukin-6, and tumor necrosis factor-alpha, which serve to galvanize fibroblasts and epithelial cells [21]. Neutrophils are either physically sloughed off or phagocytosed by macrophages [23]. The populace of inflamed cells gradually shifts to one that is dominated by monocytes over the next two to three days. These cells are then converted into macrophages, which take in bacterial leftovers and tissue while also producing numerous growth factors. The start of the expansion period of healing is contingent on the transformation of monocytes into macrophages. Macrophages release collagenases to clean the wound, in addition to IL-1 and IL-6. These macrophages also emit PDGF, TGFβ, TGFα, FGF, IGF-1, and TNFα. These growth factors and cytokines are necessary for the stimulation of keratinocytes, the recruitment and activation of fibroblasts, and the encouragement of angiogenesis [24].

#### 2.2.3. Proliferation

This phase of wound recovery includes the formation of granulation tissue, angiogenesis, and the return of structural functionality. Fibroblasts are the main cells responsible for the development of new blood vessels and collagen production in the dermis, transforming the blood clot into granulation tissue, which supplies structural and nourishing support for the recovery of the outer layers. Additionally, keratinocytes migrate and divide as they expand the newly created epithelial covering that makes up the different layers of the epidermis toward the wound’s edge [25]. The gradient in chemotactic growth factor cytokines and chemokines concentration controls the fibroblasts’ direction of migration. Two factors that influence the behavior of fibroblasts are PDGF and TGF-β. PDGF induces the multiplication, chemotaxis, and secretion of collagenase in fibroblasts, while TGF-β triggers the transcription of collagen, proteoglycan, and fibronectin genes, as well as the production of tissue inhibitors of metalloproteinases (TIMPs). As they progress, fibroblasts rearrange their shape by stretching out cytoplasmic projections to new binding sites, which they use to detach themselves from the ECM and the provisional matrix in the clot and move ahead using their cytoskeleton network. Additionally, fibroblasts manufacture and deposit collagen, proteoglycans, and other components of granulation tissue [21]. VEGF has a specific role in the production of new capillaries from existing small blood vessels, a process known as angiogenesis [21]. The body utilizes re-epithelization, a course of action spurred by inflammatory cytokines to restore a protective layer against fluid leakage and bacterial infiltration. Myofibroblasts, which are activated by macrophages during the re-epithelization process, help to contract the wound by working on actin and myosin. The consequence of cell bodies coming together to diminish the amount of tissue that requires healing is wound contraction, which reduces the size of the scar [25].

When growth factors EGF, KGF, and TGFα bind to receptors on the basal epithelial cells, these cells migrate and proliferate. This causes the desmosomes and hemidesmosomes, which connect the basal epithelial cells to their neighboring cells and basement membrane to disperse, thus creating a space between them and allowing them to move. The cuboidal basal epithelial cells transform into a flat shape and travel in a single layer across the collagen fibers of the new granulation tissue.

#### 2.2.4. Remodeling

During the remodeling phase, the granulation tissue slowly changes into scar tissue. Scar tissue has an abundance of collagen fibers and is not as cellular or full of blood vessels as regular tissue [26]. The collagen initially laid down is less dense than that of healthy skin, with its fibers running parallel to the skin. As lysine residues become more hydroxylated and glycosylated, the collagen in the granulation tissue becomes less thick. This type I collagen is thicker and follows the lines of relaxed skin tension, helping to bolster the repair tissue’s strength [24]. The wound bed includes cells that generate proteolytic enzymes such as MMPs that can destroy entire fibrillar collagen molecules, damage collagen, and degrade proteoglycans. Neutrophilelastase, a serine protease, can break down almost any type of protein. TIMPs have the ability to restrain MMPs, while α1-protease inhibitors and α2 macroglobulin can reduce serine protease activity. Fibroblasts continually produce collagen, forming larger bundles bolstered by covalent crosslinks that develop over time [21].

Despite the breakdown and production of organized collagen, as well as the addition of other proteins, collagen synthesis goes on for approximately a month.

## 3. Electrospinning Is an Emerging Way to Create Polymer Nanofibrous Structures for Aiding in Wound Healing

Electrospinning allows for the development of systems of nanosized fibers that are similar to the original structure of the ECM, which helps the cells to carry out their regular activities such as adhesion and growth [27]. The principle behind creating fibers is to employ electrostatic force, which results in the fiber being spun from the solution. In order to achieve charging potential, a high voltage is briefly applied to the fluid reservoir. As depicted in Figure 3, a syringe linked to a pump is used to insert the spinning solution, which then creates a pendant drop at the spinneret’s end with gentle pressure. The droplet is transformed into the Taylor cone, a hemispherical shape, by the action of this electric force. A steady jet can form depending on the viscosity and surface tension of the fluid. An elongated fiber structure develops on the collector when the electrostatic force is greater than the surface tension. As the solvent approaches the collector’s surface, it evaporates. Due to their high surface-to-volume ratio and tiny pores, electrospun fibrous networks can effectively cause hemostasis without the utilization of a hemostatic agent. Exudates are effectively consumed by nanofibers, which provide a soggy climate for cell development. The porosity part of these designs, with their small pores, restricts bacterial contamination, provides high penetrability, and shields injured tissue from drying out. One more key component of the electrospinning method is the capacity and adaptability to embed drugs and other bioactive synthetic substances into nanofibers, for example, development factors, anti-inflammatory agents, nanoparticles, and antimicrobials [28]. Figure 3 depicts the electrospinning process setups.

## 4. The Influence of Different Factors on Electrospinning Is Investigated

The electrospinning process is influenced by a number of factors. The factors can be divided into three groups: solution, environmental, and electrospinning parameters. Examples of the electrospinning parameters are voltage, electric field, the gap between the needle and the collector, flow rate, and needle diameter. As for the solution parameters, these include the solvent, polymer concentration, viscosity, and solution conductivity. Temperature and relative humidity are included in the environmental parameters. Table 1 represents the consequences of various electrospinning factors on fiber morphology.

### 4.1. Effect of Applied Voltage

Generally, when the voltage reaches a certain level, a current running through a metal needle from a high-voltage energy source causes a round droplet to change shape into a Taylor cone and create tiny nanofibers [29]. It is thought that the stretching of the polymer solution due to the electric repulsion within the polymer jet is what causes the production of nanofibers with a smaller diameter when the voltage is increased [30]. Utilizing PEO/water combinations, Deitzel et al. [31] reported that beads formed as the applied voltage increased. Meechaisue et al. [31] and Zong et al. [31] reported similar outcomes as well. It was concluded that the diameter of the nanofibrous scaffolds increased as the voltage increased. This was due to the jet length increasing with the higher voltage [32].

### 4.2. Effect of Solution Flow Rate

The flow of the polymeric solution through the metallic needle tip determines the morphology of the electrospun nanofibers. Beyond a certain point, increasing the flow rate results in the formation of beads in addition to an increase in pore size and fiber diameter (caused by the nanofiber jet’s failure to completely dry during its flight between the needle tip and metallic collector) [33]. For instance, when the flow rate was increased to 0.10 mL/min for polystyrene, beads began to form. However, when the rate of flow was lowered to 0.07 mL/min, nanofibers were produced with no beads. Additionally, the density of the surface charge can cause imperfections in the nanofiber structure. If the surface charge density changes, the nanofiber’s morphology might too. For instance, a direct relationship between flow rate and electric current was discovered by Theron, S et al. They investigated the effects of flow rate and surface charge density using a variety of polymers, including PEO, polyacrylic acid (PAA), polyvinyl alcohol (PVA), polyurethane, and polycaprolactone (PCL). They discovered that for PEO electrospun nanofibers can combine as they approach the collector, resulting from a boost in the flow rate, which leads to a higher electric current and a decrease in the surface charge density [34].

### 4.3. Influence of the Needle-to-Collector Distance and Needle Diameter

The morphology of an electrospun nanofiber is largely dependent on the distance between the metallic needle tip and collector. The distance between the metallic needle tip and collector varies with the polymer system in a manner similar to that of the applied electric field, viscosity, and flow rate. The distance can have a big impact on the nanofiber morphology, as it is affected by the deposition time, evaporation rate, and instability interval of the whipping process [35]. Therefore, in order to prepare uniform electrospun nanofibers, a critical distance must be maintained [36].

### 4.4. Effects of Polymer Concentration and Solution Viscosity

The combination of surface tension and an applied electric field breaks down the entangled polymer chains into smaller pieces, which prevents them from reaching the collector electrode due to the low concentration of the polymeric solution [37]. These pieces result in the development of beads or beaded nanofibers. As the concentration of the polymeric solution increases, the viscosity of the solution rises, resulting in an increase in the entanglement between the polymer chains. Surface tension is overcome by these chain entanglements, which leads to uniform electrospun nanofibers without beads. When the concentration of the solution surpasses the critical value, the flow of the liquid through the needle tip is impeded, leading to the production of beaded nanofibers.

Zong et al. observed that the form of the beads shifts as the viscosity increases while investigating PDLA and PLLA [38]. Doshi et al. also reported on the impact of concentration and viscosity on the morphology of the nanofibers. They discovered that 800–4000 pascal seconds (Pa.s) is the ideal viscosity for the creation of electrospun nanofibers while working with PEO [39].

### 4.5. Effect of Electrical Conductivity

The Taylor cone formation is influenced by electrical conductivity, which also helps control the nanofibers’ diameter. Because the surface of the droplet is not sufficiently charged to form a Taylor cone in a solution with lower conductivity, electrospinning does not occur. When the electrical conductivity of the solution reaches a critical level, the Taylor cone forms and the diameter of the fiber also decreases [40].

Conductivity stops the Taylor cone formation and electrospinning above a critical point. The electrospinning procedure begins once there are enough free charges in the conductive polymer mixture to create a Taylor cone on the fluid’s surface. Salt addition affects electrospinning in two ways: (i) it raises the ion number in the mixture, therefore growing the surface charge density of the liquid and the electrostatic power generated by the electric field, and (ii) it enhances the conductivity of the solution, thereby lessening the tangential electric field along the surface of the fluid. The effect of salt on the diameter of nanofibers has been studied by numerous researchers. For instance, KH_2_PO_4_, NaH_2_PO_4_, and NaCl in 1% *w*/*v* were studied by Zong et al. to see how they affected the diameter of polymer nanofibers (D,L-lactic acid). They observed that the nanofibers were smoother, more beaded, and had a smaller diameter than the pristine nanofibers when they added salt to the polymer solution one at a time.

### 4.6. Role of Solvent in Electrospinning

Researchers looked into the ways in which the solvent and solvent combination impacted the shape of nanofibers [41] and they came to the conclusion that the solvent has an impact on the polymer system in a manner similar to applied voltage [42]. The solvent also plays an essential role in the manufacture of highly porous nanofibers.

When a polymer is dissolved in two solvents, one of them may act as a non-solvent. The creation of highly porous electrospun nanofibers results from the phase separation caused by the different evaporation rates of the solvent and non-solvent [30]. Y. Zhang et al. [43] reported similar outcomes as well. Change in the proportions of tetrahydrofuran (THF) and dimethylformamide (DMF), according to Megelski et al. created porous nanofibers [33]. Along with the solvent’s volatility, its conductivity and dipole moment are crucial factors to consider.

Jarusuwannapoom et al. conducted an experiment involving eighteen different solvents to measure conductivity and dipole moment. Ultimately, it was determined that ethyl acetate, DMF, THF, methyl ethyl ketone, and 1,2-dichloroethane can be utilized in the electrospinning of polystyrene polymeric solution due to their comparatively higher electrical conductivity and dipole moment [44].

### 4.7. Effect of Humidity and Temperature

Recently, it was reported that environmental (ambient) factors such as relative humidity and temperature also impact the nanofibers’ diameter and morphology in addition to electrospinning and solution parameters [45]. Pelipenko et al. used PVA and PEO to study how the diameter of nanofibers changed as the humidity changed. They noticed that as humidity increased from 4% to 60%, the diameter of the nanofibers decreased from 667 nm to 161 nm (PVA) and 252 nm to 75 nm (PEO). Park and Lee also observed that the average diameter of PEO nanofibers reduces in relation to an increase in humidity, which counterbalances the effects of temperature on the diameter [32]. Temperature alters the average diameter of the nanofibers in two opposing ways. Firstly, it speeds up solvent evaporation; secondly, it reduces solution viscosity. The mean fiber diameter decreases as a consequence of both the increased dissolution and the decreased solution viscosity, which operate through two different mechanisms. Vrieze et al. reported a similar observation while working with poly(vinylpyrrolidone) (PVP) and cellulose acetate (CA) [46].

**Table 1 pharmaceuticals-16-00325-t001:** Effects of main electrospinning factors (solution, processing and ambient) on fiber morphology.

Parameters	Effect on Fiber Morphology	References
Viscosity	A thicker fiber diameter is caused by a thicker consistency of the liquid. There is no any continuous fiber formation if the viscosity is very low, and it is challenging to expel the jet from the needle tip if it is too high.	[47]
Polymer concentration	Increase in fiber diameter with an increase in concentration.	[48]
Molecular weight of polymer	Reduction in the number of beads and droplets with an increase in molecular weight.	[49]
Electrical Conductivity	Decrease in fiber diameter with an increase in conductivity.	[50]
Applied voltage	Decrease in fiber diameter with an increase in voltage.	[51]
Distance between tip and collector	Generation of beads with too small and too large distances, a minimum distance required for uniform fibers.	[52]
Feed rate/Flow rate	Decrease in fiber diameter with a decrease in flow rate, generation of beads with too high flow rate.	[53]
Humidity	High humidity results in circular pores on the fibers.	[54]
Temperature	Increase in temperature results in a decrease in fiber diameter.	[54]

## 5. Wound Dressings with Multiple Functions

Research has been conducted in recent years to develop dressings with a variety of capabilities that fulfill all the requirements for successful wound healing. Electrospinning, a scope of regular or engineered polymers and including drugs, nanoparticles, and bioactive mixtures, can be utilized to make multifunctional composite frameworks.

### 5.1. Antibacterial Activity of Electrospun Nanofibers for Wound Dressing

Antibacterial treatments for wound recuperation are a well-known research region since wound contaminations are a major worldwide concern. To limit the adverse consequences of contaminations in the injury locale, it is basic to utilize an injury dressing that can both blockade bacterial infiltration and microbial colonization into the injury site while likewise encouraging skin recovery. Most antibacterial nanofibers are fabricated by electrospinning antibacterial mixtures into the filaments. Anti-infection agents, metallic nanoparticles, and compounds obtained from regular concentrates have all been joined to work on the antibacterial properties of electrospun nanofibers. Metallic nanomaterials, for example, such as zinc oxide, silver, iron oxide, and gold nanoparticles (AgNPs), are notable for their capacity to mend wounds [55].

They can be utilized in the creation of wound dressings due to their capacity to combat human pathogens. Due to this, researchers have recently become very interested in metallic nanoparticles. Particularly fascinating are silver nanoparticles. They have a high degree of toxicity and a broad area, making them more likely to come into contact with pathogens [56]. Silver nanoparticles have been commonly used for the production of antimicrobial materials [57]. Incorporating metal nanoparticles and metal oxide into the polymeric membrane structure is one of the greatest ways to fabricate dressings with antibacterial properties. Examples of materials that have a wide pore size, great gas flow-through capability, and a high ratio of surface area to the volume include hydrogels, nanocomposites, and nanofibers. An overview of current nanoparticles and nanomaterials-based wound dressings is shown in Table 2.

The ideal approach for developing materials for wound healing appears to be the combination of hydrogels, nanocomposites, or nanofibers with nanoparticles. Multiple methods of incorporating Ag nanoparticles into the polymer structure can be achieved through electrospinning, chemical alteration, and the creation of hydrogels [58].

Bioactive wound dressings often have their antimicrobial properties increased with the addition of silver nanoparticles [59]. Silver nanoparticles offer several advantages for wound healing, such as low toxicity to the body, an antibacterial effect, and the prevention of drug resistance [60].

Ganesh et al. developed PVA–chitosan composite electrospun nanofibers with co-encapsulated Ag nanoparticles and sulfanilamide to have a synergistic wound-healing effect. In order to verify that the physicochemical characteristics of Ag nanoparticle-loaded nanofibers were successfully formulated, X-ray diffraction analysis and Fourier transform infrared spectroscopy were used. The results from SEM images indicate that the nanofibers have a continuous and smooth structure with an average diameter of 150 nm, making them suitable for encapsulating silver nanoparticles for the treatment of microbially infected wound healing. In addition, the swelling analysis showed that the amount of Ag nanoparticles and sulphanilamide had an effect on how much the PVA–chitosan nanofibers swelled. To prevent the nanofibers from absorbing water and swelling, the Ag nanoparticles and sulphanilamide were bound to the polymer matrix using hydrogen bonds [61]. Research on antibacterial activity showed that the zone of inhibition against *Staphylococcus aureus*, *Escherichia coli*, and *Pseudomonas aeruginosa* was notably increased when PVA–chitosan nanofibers were combined with Ag nanoparticles and sulphanilamide in comparison with the plain nanofiber. This implies that the antimicrobial effect of the scaffold was improved by the combination of Ag nanoparticles and sulphanilamide. In vivo results using a rat model showed that the PVA–chitosan nanofibers and the co-loaded nanofibers had similar results, both of which reached 90.76 ± 4.3% after seven days. In contrast, the control group had a wound closure rate of 55.26 ± 3.5% after 20 days. Alipour et al. developed PVA–pectin-based nanofibers with Ag nanoparticles loaded into them for use in the treatment of wounds. The creation of PVA–pectin nanofibers was confirmed by the use of energy dispersive X-ray analysis (EDS), XRD analysis, and FTIR spectra analysis. Testing of the fibers’ mechanical properties, such as elongation at break (260.5 ± 8.2%), Young’s modulus (7.7 ± 0.21 Mpa), and tensile strength (63.4 ± 3.3 Mpa), showed good results. Furthermore, the in vitro cytotoxicity evaluation of the polymeric nanofibers using the MTT assay revealed that HSF-PI 18 fibroblast cells had a high rate of viability and demonstrated strong antibacterial effects against *Escherichia coli*, *Pseudomonas aeruginosa*, and *Staphylococcus aureus strains*.

Lee et al. designed nanofibers fabricated of chitosan, with silver nanoparticles entrapped within, for the fast recovery of bacteria-infected wounds. The SEM images illustrated that the mean fiber diameters for the plain nanofibers and nanoparticle-loaded nanofibers were 460 ± 80 nm and 126 ± 28 nm, respectively. The in vitro experiments of the nanofibers demonstrated a sizeable inhibition area against Methicillin-resistant *Staphylococcus aureus* and *P. aeruginosa*, suggesting that the chitosan-based nanofibers coated with Ag nanoparticles may be a suitable treatment for contaminated wounds [62].

**Table 2 pharmaceuticals-16-00325-t002:** An overview of recent nanoparticles and nanomaterials-based wound dressings.

Electrospun Material	Nanoparticle	Bacterial Species	Composite Nanofiber Diameter	Electrospinning Parameter	Reference
The polyethylene oxide/Graphene oxide was electrospun with peppermint oil	CeO_2_	*S. aureus* and *E. coli*	310–365 nm	Electrospinning was conducted 15 cm away from the source and with a voltage of 15 kV, while the feeding rate remained at 1 mL per hour.	[63]
Nanofibrous scaffolds from Gum Arabic, polycaprolactone, and polyvinyl alcohol	Ag (10–100 nm)	*S. aureus*, *P. aeruginosa*, *E. coli*, and fungus *Candida albicans*	150–250 nm	A 10 mL syringe that held the ready-fabricated polymeric solution was filled with it, and a syringe pump was used to deliver it at a flow rate of 0.5 mL/h. The voltage that was used was 18 kV, and the distance between the tip and the collector was 150 mm.	[64]
Electrospun Chitosan/Gelatin	Fe_3_O_4_	*S. aureus* and *E. coli*	307–435 nm	The following electrospinning parameters were set: 0.8 mL/h feeding rate; 15 kV voltage; and 15 cm distance between the needle tip and collector.	[65]
Polyethylene Oxide/Carboxymethyl Chitosan Nanofibers	Ag (12–18 nm)	*S. aureus*, *P. aeruginosa*, *E. coli*, and fungus *Candida albicans*	50–300 nm	The composite fiber membranes were fabricated as follows: 40% relative humidity, a 20 cm span between needle-to-collector, and a 20 kV applied spinning voltage.	[58]
Polyvinyl alcohol–chitosan composite electrospun nanofibers	Ag	*S. aureus*, *P. aeruginosa*, and *E. coli*	150 nm	At room temperature and relative humidity of 45.5%, the electrospinning procedure was completed. The collector was placed 15 cm from the needle tip, voltage (15 kV) was used. A single syringe piston pump was used to regulate the solution feed rate at 0.5 mL/min.	[61]
Electrospun chitosan nanofibers	Ag	*S. aureus*, and *P. aeruginosa*	The average fiber diameters of CTS and CTS/AgNPs nanofibers were 460 ± 80 nm, 126 ± 28 nm, 238 ± 46 nm, 343 337 ± 49 nm and 349 ± 56 nm for AgNPs contents of 0, 4, 2, 1.3, and 0.7 wt.%, respectively	The polymer solution was put into a syringe with a Luer lock and 22 G metal blunt needle. A high-voltage DC power supply was used to electro-spin it on an aluminum foil covered rotating mandrel at 23 kV with a 1 mL/h feed rate and a needle tip-to-collector distance was 15 cm.	[62]
Ultrafine Cellulose Acetate Fibers	Ag	*S. aureus*, *E. coli*, *K. pneumoniae*, and *P. aeruginosa*	The average diameters of the cellulose acetate fibers electrospun with 0.05 and 0.5 wt.% AgNO3 were 3.3 and 6.9 nm, respectively	Distance of 10 cm from the needle tip to the ground electrode and a flow rate of 3 mL/h, CA solutions electrospun at a voltage of 17 kV.	[66]
Electrospun PVA Nanofibrous Membranes Impregnated Cellulosic Fibers	Ag	*S. aureus*	169 nm	the prepared electrospinning solution was placed in a 10 mL needle tube, and 17 G needles were installed. The spinning distance was adjusted to 15 cm. Voltage was supplied between the needle tip (+14.0 kV) and the roller collector (−3.50 kV) covered with aluminum foil. The collector speed of 80 r/min was chosen. The temperature and relative humidity were kept at (25 ± 5) °C and 30% ± 5%, respectively.	[67]

### 5.2. Electrospun Wound Dressings Loaded with Bioactive Molecules

Cells (such as neutrophils, macrophages, and fibroblasts), growth factors, and cytokines interact in a complex manner during the various stages of wound healing [68]. To improve this interaction, various biologically active molecules have been incorporated into electrospun membranes by researchers [69]. The wound-healing process needs to be managed with the precise and directed liberation of natural substances (such as growth factors, vitamins, and anti-inflammatory molecules) at the wound site [70]. Figure 4 represents the various bioactive molecules impregnated into nanofibrous scaffold and their functions in the wound-healing process.

#### 5.2.1. Growth Factors and Cytokines

GFs are physiologically active polypeptides that regulate cell proliferation, migration, differentiation, metabolism and proliferation during the healing process of wounds [71]. The recuperating system for wounds is constrained by a large number of development variables and cytokines, for example transforming development factor-β (TGF-β), fibroblast development factor (FGF), epidermal development factor (EGF), platelet inferred development factor (PDGF), and vascular endothelial development factor (VEGF) [72]. Fragments such as TGF-β, IL-1, IL-6, PDGF, EGF, and VEGF are all essential for the formation of granulation tissue, control of the inflammatory process, and the promotion of angiogenesis. In order to improve skin regeneration, Norouzi et al. [73] looked into the manufacture of core–shell nanofibrous membranes fabricated of gelatin and poly (lactic-co-glycolic) acid (PLGA) using multi-jet electrospinning. These membranes demonstrated the presence of gelatin and PLGA-EGF nanofibers with diameters that range from 315 nm to 465 nm and 130 nm to 220 nm, respectively. The swelling capacity of the membranes was improved by the combination of PLGA and gelatin nanofibers, with the swelling ratio rising from 23 ± 4% for pure PLGA fibers to 130 ± 10% for PLGA/gelatin nanofibrous membranes. In addition, after a beginning burst rescue, the EGF rescue from these membranes occurred over the course of a 9-day sustained release. Additionally, these membranes were able to increase collagen type I and III expressions, cell proliferation, adhesion, and blood clotting.

Jin, G et al. used two distinct techniques to include multiple epidermal induction factors in to gelatin/poly (l-lactic acid)-co-poly(caprolactone) (PLLCL) nanofibers: Four different types of nanofibers were prepared using the blend electrospinning approach, with diameters of 456 ± 62 nm, 382 ± 100 nm, 299 ± 46 nm, and 366 ± 125 nm. The gelatin/PLLCL/EIF (b) nanofibers had a burst release of EGF over the first three days and then stabilized, with a release of 77.8% after 15 days. On the other hand, the gelatin/PLLCL/EIF (cs) nanofibers had a more sustained diffusion of EGF, with a cumulative release of 50.9% after 15 days [74]. This controlled release had a positive effect on ADSCs, leading to a 43.6% increase in cell multiplication on the gelatin/PLLCL/EIF (cs) nanofibers compared with the gelatin/PLLCL/EIF (b) nanofibers. Furthermore, the rates of cell multiplication on the gelatin/PLLCL/EIF (cs) and gelatin/PLLCL/EIF (b) nanofibers were 560% and 404%, respectively, between days 5 and 15.

#### 5.2.2. Vitamins

Vitamins A, C, and E to the injured area can accelerate the healing process [75]. Vitamin A encourages the production of macrophages and monocytes at the wound site, as well as stimulating collagen production and re-epithelialization [76]. The ability of Vitamin E to promote angiogenesis, lessen scarring, and have antioxidant and anti-inflammatory effects also helps to speed up wound injury [77].

Sheng et al. looked at adding the PEGylated derivative of Vitamin E (TPGS), i.e., D-α-Tocopheryl polyethylene glycol 1000 succinate to Silk fibroin (SF) nanofibers in order to hasten the healing of wounds [78]. The membranes produced TPGS in an initial burst during the initial half hour after production, after which it gradually diffused over the following three days. Furthermore, the L929 cells were able to survive and proliferate on the surface of nanofibers infused with TPGS (2, 4, and 8% *w*/*w*) and SF.

In 2018, Kheradvar et al. created SF_PVA_AV nanofibers with starch nanoparticles that had been loaded with Vitamin E (VE-SNPs) [79]. The VE-SNPs generated had a mean diameter of 44.7 nm, a 91.63% encapsulation efficiency, and a round shape morphology. The SF_PVA_AV nanofibers had a diameter of 298.23 ± 6.92 nm. These nanofibers released Vit E rapidly in the first 4 h and then had a sustained release over the duration of 144 h. Moreover, the antioxidant activity increased by 34.7 ± 2.05% (for 1 mg) and 66.27 ± 3.7% (for 5 mg) due to the VE-SNPs loading. Testing showed that the electrospun membranes were biocompatible as they can help with fibroblast adhesion, spreading, and proliferation.

Vit C was incorporated into SF nanofibrous matrices by Fan and their collaborators [80]. It was found that when the quantity of Vitamin C in SF nanofibers was augmented, the average size of the fibers went up from 362 ± 121 nm (with 1 wt.% Vit C) to 416 ± 133 nm and then 506 ± 68 nm (with 3 wt.% Vitamin C).

The SF nanofibers show a burst of Vit C rescue in the initial 20 min before leveling out after 250 min (60% for 1 wt.% Vitamin C and 70% for 3 wt.% Vitamin C). The incorporation of Vit C caused the key functional genes (Col1a1, Gpx1, and Cat) in the membranes to be expressed at a higher level, resulting in more viable cells.

#### 5.2.3. Anti-Inflammatory Agents

The first anti-inflammatory molecule to be introduced into electrospun nanofibers was curcumin [81]. Curcumin can suppress the inflammatory enzymes cyclo-oxygenase (COX)-2 and lipoxygenase (LOX) as well as inhibit the release of two inflammatory cytokines (Interleukin (IL)-8 and tumor necrosis factor (TNF)-α released by monocytes and macrophages [82]. Merrell et al. conducted research on PCL nanofibers loaded with curcumin for use as wound dressings. The diameters of the nanofibers shifted from 300–400 nm (for PCL nanofibers) to 200–800 nm (for PCL/curcumin nanofibers) when 3% and 17% of curcumin were incorporated [83]. The nanofibers were capable of releasing curcumin for three days, which had a protective effect on HFF-1 cells exposed to hydrogen peroxide and decreased the pro-inflammatory response of lipopolysaccharide-stimulated mouse peritoneal macrophages. The IL-6 expression dropped from around 1220 pg/mL to around 600 and 400 pg/mL for cells treated with PCL nanofibers loaded with 3% and 17% curcumin, respectively. By day 10, mice treated with PCL nanofibers containing curcumin had nearly 80% wound closure, significantly higher than the 60% seen for mice treated with PCL nanofibers only. Due to its anti-inflammatory effects, chrysin, a natural flavonoid found in a variety of plant extracts have also been added to nanofiber-based wound dressings [84]. Deldar and his team blended chrysin into PCL/poly(ethylene glycol) (PEG) nanofibrous meshes to create a wound dressing with anti-inflammatory and antioxidant characteristics. This is because chrysin is known to impede nitric oxide (NO) synthase, the manufacture of NO, and the release of TNF-α and IL-1β, as well as reducing lipopolysaccharide-activated COX-2 expression [85].

Mohiti-Asli and their fellow researchers incorporated ibuprofen into PLA nanofibers to promote the mending of third-degree wounds [86]. The presence of ibuprofen in the production of nanoscaffolds led to an increase in their diameters, with values of 329.11 ± 249.62 nm, 478.31 ± 167.61 nm, and 585.38 ± 131.51 nm for 10%, 20%, and 30% of ibuprofen, respectively. These nanofibrous mats were noted to be helpful in facilitating cell attachment, especially the PLA nanofibrous mat containing 20% ibuprofen. In addition, the PLA nanofibrous mat with 20% IBP was found to be capable of degrading when applied to third-degree wounds in mice, resulting in a 60% wound contraction after 14 days. Table 3 outlines a selection of growth factors, vitamins, and anti-inflammatory agents that can be included in electrospun meshes for wound healing.

## 6. Wound Healing Using Bio-Based Electrospun Fibers

Bio-based polymers are regular macromolecules fabricated by carrying on with living things (generally called biopolymers). Plants (cellulose, lignin), animals (collagen, chitin, chitosan), microorganisms (bacterial cellulose, PHA), and biotechnological processes are wellsprings of biopolymers. By virtue of their specific credits, they have displayed promising results in biomedical applications; for instance, drug movement, tissue planning, and wound patching are listed in Figure 5.

Biopolymers, in their unadulterated structure or in a mix with different polymers, can be framed into stringy platforms, making them an engaging contender for skin substitution. Various electrospun biopolymer dressings with different functions and wound type targets are listed in Table 4.

### 6.1. Biomaterials for Wound Healing: Cellulose Electrospun Nanofibers

Cellulose, is a naturally occurring biopolymer that is friendly to the environment and biocompatible and biodegradable, and has multiple biomedical uses such as constructing scaffolds for tissue regeneration, wound dressings, artificial tissue and skin, managing drug release, cleaning blood, and creating materials for cell cultures [96]. Similar research was performed on bacterial cellulose-based scaffolds to see if they can be used in pre-clinical and clinical trials, such as for wound dressings for skin lesions [97]. A patient with second-degree burns on his face was provided Nanocell®, a bacterial cellulose-based scaffold, in order to facilitate the healing process without the need to apply additional bandages to the wound sites. After two weeks, the facial burns were completely repaired without any allergic or skin-irritating reactions, demonstrating the effectiveness of bacterial cellulose dressings for treating burned skin. Electrospinning is a novel technology that can be utilized to produce cellulose nanofibers, various polymer/cellulose mixtures, or blends of cellulose with nanoparticles that possess increased functional qualities, such as antimicrobial features to prevent wound site infection [98].

Nonwoven nanofibers with extensive surface areas and closely interlinked pores are especially useful for treating wounds as they are able to absorb large quantities of exudate and promote effective gas exchange [99]. Additionally, cellulose scaffolds can carry a range of bioactive compounds, such as molecules that reduce inflammation and kill bacteria [100]. Roy et al. [101] confirmed the potency of bamboo-incorporated paclitaxel-infused cellulose electrospun fibers in the treatment of skin cancer and wound recovery. Song et al. conducted an experiment where the surfaces of cellulose, carboxymethylated cellulose (CMC), and ribbon-shaped CA electrospun fibers were modified with Ag nanoparticles at various pH levels [102]. The antimicrobial characteristics and application potential of CMC fibers were amplified by the addition of silver nanoparticles which were arranged in the following order at the same pH levels: CMC > cellulose > CA.

### 6.2. Chitosan Electrospun Nanofibers for Wound Healing

Chitin and its deacetylated version, chitosan, have antibacterial, biocompatible, and hemostatic characteristics, which can be used to heal injuries [103]. Chitosan has antibacterial action in its capacity as a feeble polybase because of the presence of countless amino gatherings on its chain. The favourable to incendiary properties of chitosan have been proposed to assume a significant part in injury recuperating. Chitosan initiates macrophages, which helps with the recuperating of wounds. Moreover, chitosan can cause polymorphonuclear neutrophils (PMNs) to relocate during the beginning phases of wound mending, bringing about the arrangement of granulation tissue. Jayakumar et al. showed that chitosan has the capacity to stimulate the regeneration of the skin’s granular layer and re-epithelialization [104].

Furthermore, due to its strong bond with the injury and its capability to interact with negatively charged red blood cells, chitosan is able to successfully halt bleeding [105]. Min et al. fabricated chitin and chitosan nanoscaffolds for wound dressing by using electrospinning with 1,1,1,3,3,3-hexafluoro-2-propanol as the spinning solvent. The pro-inflammatory properties of chitosan have been recommended to assume a significant part in injury recuperating. Chitosan enacts macrophages, which support the mending of wounds. Moreover, chitosan can cause polymorphonuclear neutrophils (PMNs) to relocate during the beginning phases of wound mending, bringing about the development of granulation tissue.

Chitosan compounds containing quaternary ammonium bunches have been demonstrated to be antibacterial and antifungal. The cytoplasmic film of the bacterial cell is normally perceived as the objective area of these cationic polymers. Wound dressings can be produced using miniature and nanoscale nanofibrous materials. Chen and partners portrayed a nanofibrous layer fabricated of chitosan and collagen that has been displayed to work on injury recuperation. The film was found to advance injury recuperation, cell relocation, and expansion. Noh et al. investigated the cytocompatibility of chitin nanofibers.

The three-layered properties of these materials and their high surface area to volume ratios may be responsible for this, as they are perfect for cell attachment, advancement, and multiplication. In one review, to fabricate composite fibers, PEO/chitosan arrangements and polycaprolactone (PCL)/olive oil arrangements were electrospun together. Olive oil, chitosan, PCL, and PEO composite nanofibers were successfully created using the electrospinning process. By employing SEM and FTIR, the structure and shape of electrospun nanofibers were determined. The response surface methodology and Box-Behnken design were utilized to identify the relationship between the parameters of the process and the diameter of electrospun nanofibers, with the predicted minimum value of diameter being 88 nm when voltage, TCD, and flow rate were used at three levels (21.2 kV, flowrate of 0.2 mL hr^−1^, and tip to collector distance of 14.3 cm). These data were consistent with the experimental data of 86 nm. It was established that the optimal weight percentage of olive oil was 2% through the integration of different weight percentages of olive oil into electrospun nanofibers. After a one-day study, 58.1% of the total olive oil encapsulation was released, according to the in vitro release behavior of olive oil from PEO/chitosan/PCL/olive oil scaffolds. It was observed that cell attachment of the created nanofibrous scaffolds was satisfactory, with cell proliferation and a non-toxic behavior being demonstrated by the cytotoxicity results [106].

### 6.3. Effectiveness of Electrospun PLA Nanofibrous Scaffolds in Wound Healing

Polylactic acid (PLA) is a biopolymer that is created from a mixture of synthetic elements. Its components, such as maize starch, sugar, and rice are easy to obtain and are used in the process of ring-opening or lactic acid condensation polymerization to form the PLA monomer, which is 2-hydroxypropionic acid. There are three isomers of PLA: poly(d-lactide), racemic poly(dl-lactide), and poly(l-lactide) [107]. PLA is a thermoplastic polyester with thermomechanical properties that is both biocompatible and biodegradable, as well as able to be absorbed by living organisms. Alves et al. demonstrated that PLA electrospun membranes can be used effectively as drug delivery systems for sustained release applications in the formation of wound dressings [108]. The efficiency of dexamethasone acetate (DEX) and betamethasone being released was tested using physical adsorption and mix electrospinning techniques through PLA electrospun strands. The effects of the drugs on the morphological and mechanical properties of the PLA fibers were investigated. When the pharmaceuticals were mixed with the PLA solution and electrospun, the drug-loaded fibers had a more consistent release profile over the initial five hours than when the drugs were simply attached to the PLA electrospun membranes. Moradkhannejhad et al. investigated how PLA electrospun nanofibers loaded with curcumin can have their hydrophobicity changed by adding PEG with different molecular weights and concentrations [109].

Yang, C et al. employed coaxial electrospinning to create a (PGS)/PLLA fibrous scaffold with a PGS core and a PLLA shell. This type of structure showed enhanced cell proliferation and less inflammation than a pure PLLA scaffold. Additionally, when the poly glycerol sebacate or CUR or PEG nanofibers were incubated in PBS, an increase in weight loss values was observed when both the concentration and molecular weight of PEG were adjusted. The electrospun CUR-loaded PEG or PLA membranes were able to adjust the hydrophilicity and hydrophobicity of the medium, thus providing favourable conditions for cell growth as well as improved drug delivery. The porous shell surface shape of the generated fibers highlighted their exceptional ability to repair skin injured tissues [107]. Augustine et al. used coaxial electrospinning to create wound-healing nanofibers that consisted of core–shell fibers. Yang, C et al. utilized this method to create a core–shell structured PLLA/chitosan nanofibrous scaffold. GO nanosheets were then coated on the core PLLA-shell chitosan nanofibers to create a synergistic microenvironment for fast wound recovery [110]. This increased the membrane’s hydrophilicity. Chitosan/PLLA nanofibrous scaffolds with GO coating showed promising antibacterial activity and stimulated the growth of pig iliac endothelial cells. In rats, GO-coated chitosan/PLLA nanoscaffolds provided a positive effect on wound healing [111].

### 6.4. Utilization of Electrospun PHA Nanofibrous Scaffolds in Wound Healing

PHAs are a type of biopolyester thermoplastic [112]. Many bacteria create them as a substitute source of nutrition (carbon) when growth is not steady.

Shishatskaya et al. [113] used poly (3-hydroxybutyrate-co-4-hydroxybutyrate) due to its low crystallinity and good elasticity, which is considered one of the best PHAs for producing electrospun fibers for wound-healing applications. They found that the presence of fibroblasts had a significant influence on the quantity of hyperemia and purulent exudate, and concluded that composite fibers were a better choice for wound-healing applications. It was determined that wound healing under the cell-loaded poly (3-hydroxybutyrate-co-4-hydroxybutyrate) membrane was 1.4 times faster than wound healing under the cell-free membrane and 3.5 times faster than wound healing under the eschar membrane (control) [114]. Over the last three decades, PHA has been shown in numerous studies to have a variety of benefits over other biomaterials in medical applications, including biocompatibility, mechanical stability, strength, and biodegradability under physiological conditions with non-toxic degradation products. It was discovered that some PHA breakdown products have potential use in the pharmaceutical industry and have demonstrated a growth-suppressing effect on bacteria [115].

The cell-loaded membrane group regained all its areas after two weeks, while the pure poly (3-hydroxybutyrate-co-4-hydroxybutyrate) meshes in the control groups had 90% and 70% area reductions, respectively.

The application of electrospun fibrous meshes containing keratin and 3-hydroxybutyrate-co-3-hydroxyvalerate (PHBV) for wound dressing purposes was investigated [116]. Azimi et al. used electrospinning to manufacture a combination of poly (3-hydroxybutyrate)/poly(3-hydroxyoctanoate-co-3-hydroxydecanoate) [P(3HB)/P(3HO-co-3HD)]. Additionally, they decorated the fiber meshes with chi-tin-lignin/glycyrrhizin acid (CLA) complexes through electrospray [117]. CLA complexes are organic molecules on a microscopic scale that have been demonstrated to be useful for enhancing the skin’s properties when tested on an artificial skin sample [118].

Kandhasamy et al. designed a composite scaffold with a PHB/gelatin/ostholamide blend coating of collagen, which displayed strong anti-inflammatory action, and which is an encouraging indication for wound relief applications. The incorporation of ostholamide into the scaffold provided increased mechanical stability, a slow enzymatic breakdown rate, and successful anti-bacterial action against *Pseudomonas aeruginosa and Staphylococcus aureus* [119]. The NIH 3T3 fibroblast proliferation experiments in both in vivo and in vitro settings proved to be very compatible as seen in successful wound relief in Wistar rats. After fifteen days, the PHB or gelatin or ostholamide (OSA) or collagen scaffold had entirely closed the wound, while the PHB or gelatin or OSA or collagen scaffold, the pure collagen scaffold, and the cotton gauze positive control had all reduced the wound size by 75%, 65%, and 45%, respectively.

### 6.5. Evaluation of Electrospun PCL Nanofibrous Scaffolds in Wound Healing

Polycaprolactone (PCL) is the most in-demand material due to its semi-crystalline nature and its ability to be adjusted in terms of mechanical properties and solubility in a variety of solvents. This makes it a great option for combining with other polymers. Additionally, compared with other polyesters, PCL breaks down at a slower rate, making it beneficial for certain purposes [120]. The degradation products of PCL are naturally non-toxic, making it a suitable component for skin regeneration therapies. It is often combined with natural polymers such as collagen before electrospinning, or collagen is deposited on the PCL nanofibers. Additionally, PCL has been mixed with gelatin, derived from collagen, and gelatin has been used as the core polymer in core–shell PCL/gelatin nanofibers. Double-nozzle electrospinning has been used to produce separate gelatin and PCL nanofibers, generating two distinct types of nanofibers which can be mixed or layered in the scaffolds [121]. Compared with the other structures, multilayered and blended ones were discovered to be most suitable for the majority of native skin needs. These include whey keratin, protein, chitosan, hyaluronic acid, and fibrinogen [122]. Gum Arabic and Zein (a corn protein) are both utilized as food additives [123]. In addition to PCL, other natural polymers were used to modify the nanofibers. The electrospinning solution combined PCL with these organic polymers, and growth factors such as epidermal growth factor were immobilized onto PCL/collagen or PCL/gelatin nanofibers [124]. TGF-β1 has been incorporated into PCL/collagen electrospinning solutions in the past to alter PCL nanofibrous scaffolds. In this study, alaptide or l-arginine was included in PCL electrospun nanofibrous membranes. Back in the 1980s, Kasafirek et al. at the Research Institute for Pharmacy and Biochemistry in Prague, Czechoslovakia, synthesized alaptide, a spirocyclic dipeptide, which was created to act as an analogue of the melanocyte-stimulating hormone release-inhibiting factor [125].

Alaptide showed considerable promise for enhancing transdermal drug absorption and for repairing harmed tissue [126]. Arginine acts as a precursor for nitric oxide, which is correlated with wound healing. This substance aids the growth, resistance to cell death, and immunity of fibroblasts, which are integral to the healing of wounds [127] and promoted the healing of wounds by forming new skin cells and increasing blood flow.

### 6.6. Utilization of PES Electrospun Fibers in Wound Healing

Kanji et al. [128] demonstrated that human umbilical cord blood-derived CD34+ cells expanded on Polyether sulfone nanofibers can be effectively used to treat diabetic wounds, speeding up wound closure and improving re-epithelialization and neovascularization. Additionally, the use of CD34+ cells was found to reduce the pro-inflammatory activity of NF-κB and TNF-α, IL-1β, and IL-6, and to prevent MMP-1 from being expressed at a high level. It appears that PES nanofibrous membranes created using electrospinning can be employed as a fresh approach to wound healing, as they have higher absorption capabilities, more matured fibroblast production, and more collagen deposition than other commercial wound treatments.

### 6.7. Wound Healing Using PS Electrospun Fibers

In its non-crystalline form, polystyrene is a transparent and colorless solid that is fragile, rigid, and has high electrical insulation and minimal dielectric loss. To create a wound dressing, polystyrene was spun with poly caprolactone and chamomile extract, which contains phenolics and flavonoids such as apigenin which is known to have strong wound-healing capabilities [129]. Skin cells were grown in bioreactors that had motion and on electrospun polystyrene nanofibrous scaffolds which were situated at the air/liquid interface [130].

### 6.8. Application of PAA Electrospun Fibers in Wound Healing

Using poly(acrylic acid) nanofibers combined with reduced graphene oxide, photothermal activation of the nanofibers was used to modulate antibiotic distribution [131]. Another study concluded that electrospun nanofibers fabricated of poly(acrylic acid) and a synthetic biodegradable elastomer called poly(1,8-octanediol-co-citric acid) had intrinsic antibacterial activity and could be used to deliver physiologically relevant growth factor concentrations topically [132].

### 6.9. Application of Thermoresponsive Electrospun Fibers in Wound Healing

The physical and chemical characteristics of the thermoresponsive polymer used in the fabrication of scaffolds combined with the biological host’s regulatory mechanisms are influential in the behavior and function of the scaffolds. Taking advantage of these biological reactions, these scaffolds and the biomaterials that compose them can be used to create novel drug delivery systems that can be used to hasten the wound-healing process, manage the scarring process, and moderate the inflammatory response [133]. Bioactive thermoresponsive nanofiber mats have been created to aid in the treatment of wounds. Various polyblend nanofiber formulations are formed by combining PNIPAAm, PCL, and egg albumin with various concentrations of gatifloxacin hydrochloride (0–20%). These nanofibers exhibit rapid drug release within the first 10 h and sustained release over 696 h. In order to test the antibacterial capabilities of these gati-loaded nanofibers, they are evaluated against the Gram-positive pathogen *Staphylococcus aureus*, which is commonly present in wounds. In the presence of Gati-loaded scaffolds, a decrease in bacterial growth is evident, and the antibacterial properties become more potent as the concentration of Gati rises. In a study involving a rat model, the capacity of nanofibers to stimulate wound healing in vivo was evaluated. The wounds treated with 15% Gati-loaded nanofibers experienced significantly rapid healing than the wounds in control; after 21 days, 95% of the wounds had healed, while only 45% of the wounds in control had been healed. The results of this study demonstrate the capability of thermoresponsive scaffolds in wound healing. Karri et al. [134] created a new nanohybrid scaffold by combining curcumin with chitosan nanoparticles and then embedding it into a collagen scaffold to enhance tissue expansion. According to this study, a combination of curcumin, chitosan, and collagen created using coaxial electrospinning showed excellent healing in diabetes. This combination of drugs, known as a core–shell nanofibrous bioactive insulin-loaded PLGA scaffold created by Lee et al., was able to release a synthetic hormone over a four-week period, aiding in the healing of diabetic wounds [135]. Table 5 shows the various electrospun scaffolds used in the healing of wounds.

## 7. Current Commercial Electrospun Wound Dressing

Pathon, TPP-fibers (TecophilicTM), SurgiCLOT, and SpinCareTM are all products that can be used to dress a wound. Table 6 outlines the various commercial electrospun products that can be employed for wound healing. SpinCareTM is a portable electrospinning tool that is often utilized for wound management. SurgiCLOT is a bio-based and polysaccharide wound dressing that is derived from dextran [138]. Glucose-connected polymers encompass dextran, which can be extracted from sucrose through the use of particular lactic acid bacteria such as *Lactobacillus* spp., *Leuconostoc mesenteroides*, and *Streptococcus mutans* [139]. Due to dextran’s resistance to both cell adhesion and protein adsorption, hydrogels fabricated from this polysaccharide make excellent scaffolds for soft tissue engineering applications. This biopolymer, which is sometimes referred to as a type of synthetic polymer called polyurethane, is used in clinics because of its antiplatelet characteristics. However, one unique quality of polyurethanes is that they can have natural sources for their monomers. Vegetable oils, such as canola, castor, and olive oils, are the main source of polyols in nature. Therefore, it is possible to anticipate that green polyurethanes will eventually be used in biomedical products. The chemistry of the polyurethane and the method of degradation both have a notable effect on the resulting by-products. Three distinct processes involved in the biological degradation of polyurethanes in the natural environment: hydrolytic, enzymatic, and oxidative.

## 8. Chronic Diabetic Wound Healing Based on Electrospun Nanofibers

Augustine and other researchers presented the formation of a novel cerium oxide nanoparticle composed of an electrospun poly (3-hydroxybutyrate-co-3-hydroxy valerate) membrane [144]. Wound-healing studies in diabetic rats concluded that poly (3-hydroxybutyrate-co-3-hydroxy valerate membranes combined with 1% cerium oxide nanoparticle were completely compatible with cells, making them potential biomaterials for treating the diabetic wound. Lee et al. developed core–shell nanofiber scaffolds that incorporated polylactic glycolic acid and insulin solutions, allowing for a managed release of insulin for four weeks [135]. These scaffolds had more hydrophilic properties than blended nanofibrous scaffolds and were able to retain more water and found to improve diabetic wound, decrease the quantity of type I collagen in vitro, and increase the amount of transforming growth factor-beta (TGF-β) in vivo. Chen et al. created composite nanofiber, containing Polyvinyl alcohol and Chitosan. This composite also had hemostatic and anti-bacterial qualities and greater mechanical strength [91]. A three-layer synthetic scaffold was created, with chitosan as the first layer, a combination of chitosan and polyvinyl alcohol as the second layer electrospun, and nanobioglass (with up to 40% PVA) as the third layer electrospun. These composite nanofiber mats showed great cytocompatibility with fibroblasts in vitro. On its own, Polyvinyl alcohol can lower the pH of a wound site (5.8–6.2), which can slow down cell growth. However, the addition of chitosan helps to keep the pH level around 6.5, allowing for faster wound healing with less cell damage. In diabetic mice, the resulting composite nanofiber membrane significantly boosted the wound closure rate by reducing inflammation, neovascularization, and increasing the collagen synthesis. The combination of poly (lactic-co-glycolic acid) and collagen in hexafluoro-2-propanol as the solvent enabled the electrospinning technique to form nanofibers. Moreover, this created a humid atmosphere which encouraged cell movement. Research in diabetic mice has demonstrated that metformin can activate the AMPK/eNOS pathway by enhancing the angiogenesis of endothelial progenitor cells [145]. When the PLGA-collagen nanofiber is blended, it brings about a diminishment in the fiber diameter, a rise in the water contact angle, and a higher absorption of water by the nanofibers. Animals with diabetes that had the PLGA-collagen-Met nanofiber treatment showed almost 95% closure of the wound in comparison with those only receiving gauze treatment, which had 73% wound closure [146]. Kargozar et al. examined that the combination of gum tragacanth, poly caprolactone, and polyvinyl alcohol created nanofiber mats with a ninefold higher tensile strength than mats composed of polycaprolactone and gum tragacanth alone, reaching 2.7 MPa in comparison with 0.3 MPa [147]. A rise in the amount of polyvinyl alcohol and gum tragacanth caused a decrease in nanofiber diameter, leading to improved cytocompatibility and accelerated healing of wounds in diabetic rats due to a decrease in polycaprolactone hydrophobicity.

## 9. Conclusion and Future Prospectives

Nanofibers-based wound dressings have more potential and effectiveness than conventional wound dressings. The wound-healing process is considered to be incredibly complex since it is crucial for maintaining homeostasis. Biomimetic dressings can be fabricated by electrospinning with improved biological activity to promote tissue regeneration, but only a limited number of materials have proved suitable. An emerging field of study reveals the cellular and molecular control mechanisms that regulate the inflammatory response during wound healing, providing an important contribution to pathological tissue repair, as well as theoretical support for electrospinning-based wound-healing regulation. The use of electrospun materials, drugs, or cells to directly stimulate physiological repair has become a promising research field at the molecular level. It may be technically challenging to do so on a molecular level, but it seems possible to mimic the ECM by combining the biological properties of natural polymeric materials with nanoscale structures, thereby stimulating cell migration and proliferation, controlling inflammation, and speeding up wound healing. Electrospun membranes help with cell adhesion, proliferation, and differentiation, and can help prevent or decrease skin infections. In the future, wound-healing scaffolds include theranostic materials that combine interactive and bioactive means together with therapeutic and diagnostic functionality into a single scaffold. It is envisioned that new technologies integrate target biomarkers into scaffolds to monitor wound healing. In addition to excellent antimicrobial, angiogenic, antiproliferative, anti-inflammatory properties, and nanofibers impregnated with biological macromolecules also exhibit greater biocompatibility and biodegradability, as well as high surface-to-volume ratios. However, there are few shortcomings of electrospun nanofibers in the field of wound healing. Therefore, the future challenges can be addressed as follows: (i) By using multiaxial electrospinning setups with multiple needles/needleless spinning techniques, the yield of nanofiber fabrication can be enhanced in industrial and large-scale settings. (ii) Biocompatible nanofibers are required to obtain the best possible morphology, composition, pore size, and diameter for efficient market-scale technology transfer. (iii) Solving the environmental issues and safety matters in solvent evaporation during the spinning process: solvent recovery systems or green chemistry techniques for melt electrospinning. In the near future, the combination of multiple nanofiber production processes and surface modification strategies, such as heat and plasma treatment, will allow the nanofiber’s physiochemical properties to be improved. Furthermore, the creation of pH, temperature, light, electrical, or magnetic field responsive nanofibers will allow for the regulated or multi-stage release of biological molecules at the wound site. Clinical trials must be conducted in order to commercialize drug delivery systems based on electrospun membranes intended for skin regeneration and to enhance the quality of life for patients.

## Figures and Tables

**Figure 1 pharmaceuticals-16-00325-f001:**
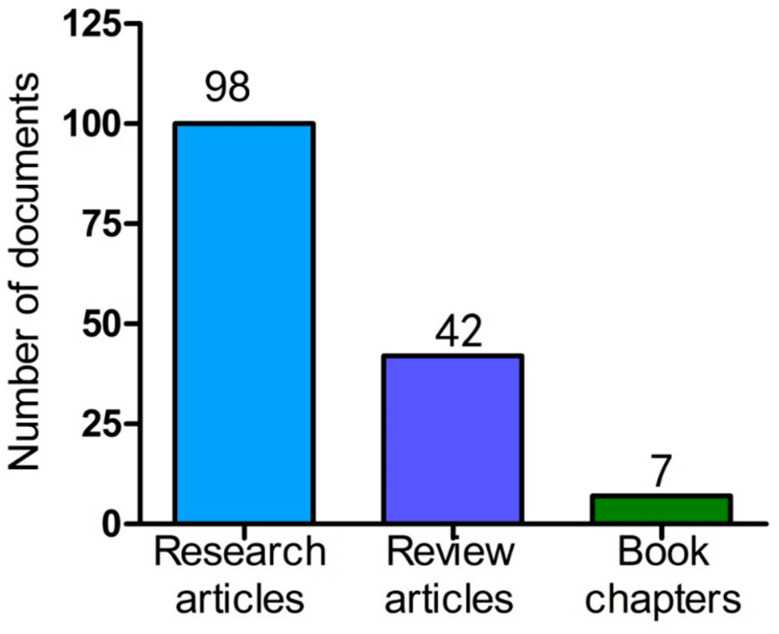
Total number of documents screened for review manuscript preparation. Documents search has been performed using keywords such as nanofibers, polymers, scaffolds, wound healing, electrospinning, nanoscaffold, and biodegradable polymer.

**Figure 2 pharmaceuticals-16-00325-f002:**
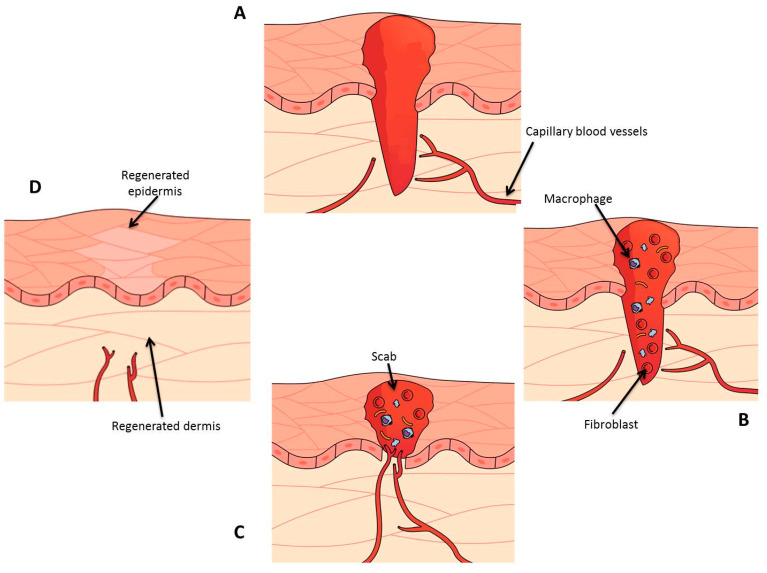
This figure depicts the four stages of wound healing, which are (**A**) hemostasis, (**B**) inflammation, (**C**) proliferation, and (**D**) extracellular matrix remodeling. During wound healing, fibroblasts (the body’s connective tissue cells) and macrophages, which defend against infection, play an enormous role.

**Figure 3 pharmaceuticals-16-00325-f003:**
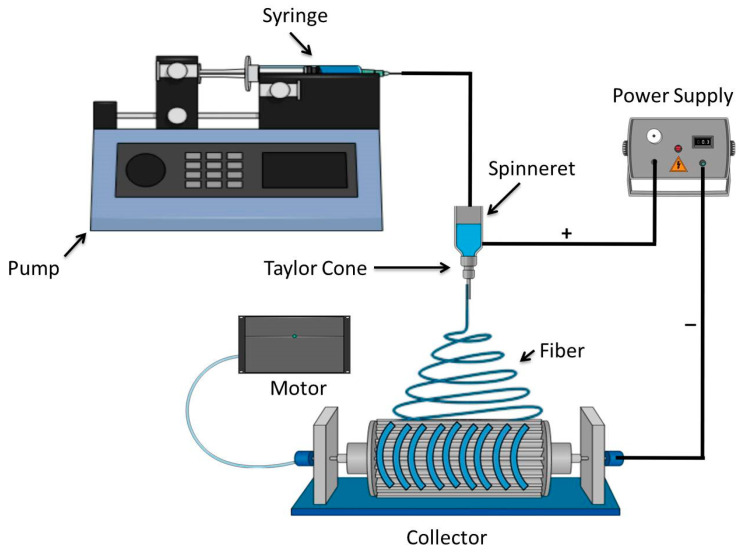
Electrospinning process setups.

**Figure 4 pharmaceuticals-16-00325-f004:**
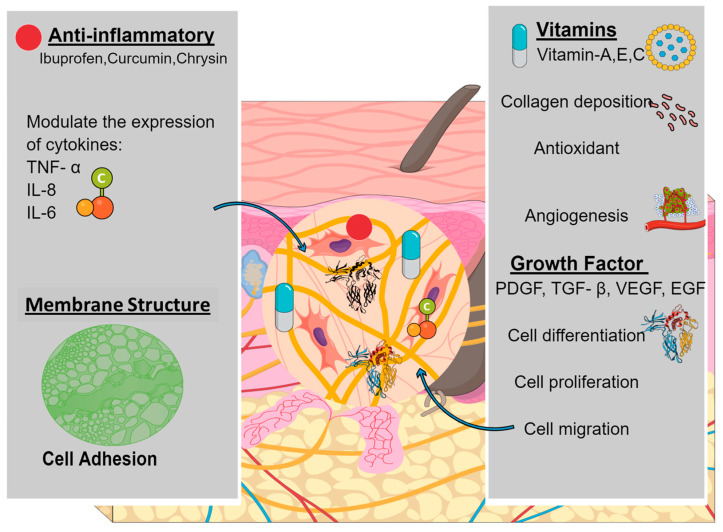
Represents the various bioactive molecules impregnated into nanofibrous scaffolds and their function in wound-healing process.

**Figure 5 pharmaceuticals-16-00325-f005:**
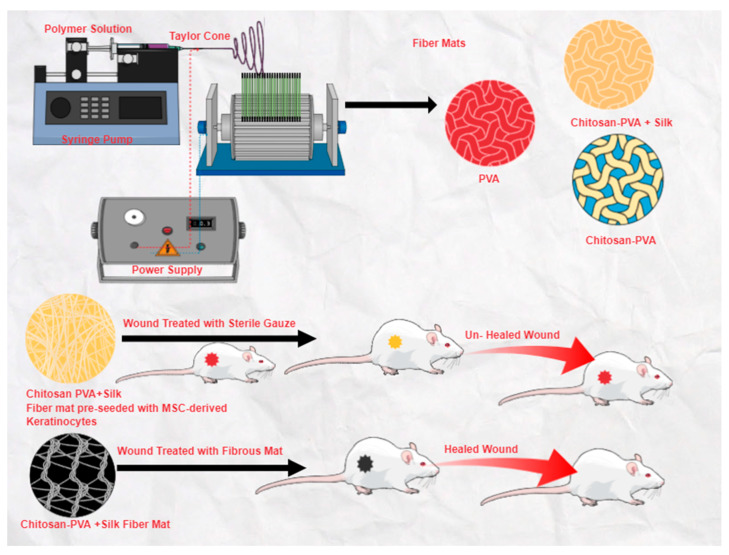
Electrospun Chitosan–PVA-Silk nanofibers mat for wound healing.

**Table 3 pharmaceuticals-16-00325-t003:** List of different growth hormones, vitamins, and anti-inflammatory agents that can be integrated into electrospun fabrics for treating wounds.

Growth Factor	Polymer	Technique Used	Fiber Diameter	Release Profile	Main Findings	Reference
Epidermal growth factor (EGF)	Gelatin and a combination of lactic acid and glycolic acid (PLGA)	Emulsion electrospinning	Diameter of Polylactic glycolic acid: 630 ± 80 nm Diameter of Gelatin: 175 ± 45 nm Diameter of Polylactic glycolic acid /Epidermal growth factor/gelatin: 390 ± 75 nm	The EGF release profile shows a rapid release on day 1, and continuous release throughout the course of nine days (total EGF release was 4.2 ± 0.2 ng/mL).	The MTT experiment revealed that fi fibroblast proliferation was greater on PLGA/EGF/gelatin membranes.	[73]
EGF, insulin, hydrocortisone, and retinoic acid	Gelatin/ PLLCL	Blend electrospinning: gelatin/PLLCL (b) Co-axial electrospinning: gelatin/PLLCL (cs)	Diameter of PLLCL: 456 ± 62 nm Diameter of Gelatin/PLLCL: 382 ± 100 nm Diameter of gelatin/PLLCL/EIF (b): 299 ± 46 nm Diameter of gelatin/PLLCL/EIF (cs): 366 ± 125 nm	Gelatin/PLLCL/EIF (b): approximately 77.8% of Epidermal Growth Factor had been released after 15 days. Gelatin/PLLCL/EIF (cs): approximately 50.9% of EGF released after 15 days.	The ASCs (adipose-derived stem cells) multiplied to a greater extent (43.6%) when in a combination of gelatin/PLLCL/EIF (cs) than when in gelatin/PLLCL/EIF alone. (b) The proportion of differentiated epidermal cells was 62.2% and 43.0% for gelatin/PLLCL/EIF (cs) and gelatin/PLLCL/EIF (b) membranes, respectively.	[74]
Vitamins	Polymer	Technique used	Fiber diameter	Encapsulation efficiency	Main conclusions	Reference
Vitamin E	SF	Blend electrospinning	Silk Fibroin/Vit E (2%) average diameter: 722.52 ± 300 nm Silk Fibroin/Vit E (4%) average diameter: 387.9 ± 114 nm Silk Fibroin/Vit E (8%) average diameter: 384.3 ± 161 nm	Silk fibroin/Vitamin E (2%): 65.82%. Silk fibroin/Vitamin E (4%): 78.59%. Silk fibroin/Vitamin E (8%): 70.02%.	In the initial 30 min, there was a rapid release of the drug observed from Vit E, which was then followed by a gradual release over the next 72 h. Moreover, supplementing L929 cells with Vitamin E allowed them to adhere better and grow faster.	[78]
Vitamin E	SF/PVA/AV	Vit E loaded into starch nanoparticles and blend with electrospinning solution	SF/PVA/AV: 298.23 ± 6.92 nm	Vitamin E into starch nanoparticles was 91.63%.	Vitamin E released quickly at first, then slowly over the course of 144 h; The higher Vit E level led to enhanced antioxidant activity. In contact with electrospun membranes, the fibroblasts cells continued to be functional, adhering and dividing.	[79]
Vit C	Silk Fibroin	Blend electrospinning	Silk Fibroin: 362 ± 121 nm Silk fibroin/Vitamin C (1%): 416 ± 133 nm Silk Fibroin/Vitamin C (3%): 506 ± 68 nm	NA	A burst peak was visible in the Vit C release profile from SF nanofibers during the first 20 min. Vitamin C had a positive effect on the viability of fibroblasts, as well as boosting the mRNA levels of key genes such as Col1a1, Gpx1, and Cat.	[80]
Anti-inflammatory molecule	Polymer	Technique used	Fiber diameter	Release profile	Main findings	Reference
Curcumin	Polycaprolactone	Blend electrospinning	Polycaprolactone: 300–400 nm Polycaprolactone/curcumin: 200–800 nm	Polycaprolactone or curcumin (17%): 35 microgram was released on 3rd day. PCL/curcumin (3%): 20 microgram was released on 3rd day.	The antioxidant qualities of fibers loaded with curcumin were further demonstrated by the Oxygen Radical Absorbance Capacity (ORAC) assay. The membranes had a cytoprotective effect on human fibroblast cells and were biocompatible. In vivo tests conducted in a living organism demonstrated that PCL/curcumin nanofibers had the capacity to accelerate the recovery of injuries in a diabetic mouse model.	[83]
Chrysin (Chr)	PCL/PEG	Blend electrospinning	Polycaprolactone/poly (ethylene glycol): 300–400 nm Polycaprolactone/poly (ethylene glycol)/Chrysin: 250–650	Polycaprolactone/poly(ethylene glycol)/Chrysin (5%): 25 micrograms of Chrysin was released after 3 days. Polycaprolactone/poly (ethylene glycol/Chrysin (15%): 96%): 43 micrograms of Chrysin was released after 3 days.	It was demonstrated that the nanofibers, which had antioxidant properties and were biocompatible, had cytoprotective effects on human fibroblast cells when they were loaded with Chrysin. The reduced levels of IL-6, IL-1β, tumor necrosis factor-α, and NO in macrophages showed the anti-inflammatory capabilities of Chrysin-filled nanofibrous mats.	[85]
Ibuprofen	PLA	Blend electrospinning	Poly lactic acid/ibuprofen (10%): 329.11± 249.62 nm Poly lactic acid/ibuprofen (20%): 478.31 ± 167.61 nm Poly lactic acid/ibuprofen (30%): 585.38 ± 131.51 nm	0.25 milligrams of ibuprofen was released from poly lactic acid/ibuprofen (30%) nanofibers after 336 h.	The nanofibers with 20 wt.% IBP had the highest cell viability and proliferation. The addition of IBP to PLA nanofibers encouraged the growth and viability of both HEK and HDF.	[87]

**Table 4 pharmaceuticals-16-00325-t004:** Different electrospun biopolymer dressings with different functions and wound type targets.

Electrospun Mesh	Incorporated Therapeutics	Function and Wound Type	References
Chitosan/Poly (l-lactide)	Graphene oxide	Antimicrobial action in infected chronic injuries.	[88]
Chitosan/keratin/polycaprolactone	Aloe vera extract	Burn and acute wounds can be aided by properties that are anti-inflammatory, antibacterial, antiviral, and antioxidant.	[7]
Polyhydroxyalkanoates	Dodecyl trimethylammonium chloride biocide	The antioxidant, anti-inflammatory, and anti-infective qualities of certain substances can provide a boost to cell reinforcement and angiogenic properties for diabetic injuries, resulting in antimicrobial effects for chronic wounds.	[89]
Polydopamine or polylactic glycolic acid	Fibroblast growth factor and ponericin G1 are both present	The skin tissue regeneration process has antibacterial and cell growth-promoting properties.	[90]
Chitosan/Polyvinyl alcohol	Nanobioglass	For chronic injuries, biocompatibility, antimicrobial action, and recovery advancement.	[91]
PLGA	Ciprofloxacin	Antibacterial and skin tissue regenerative effects that encourage cell growth	[90]
PLA	Doxycycline	Antibacterial activity, chronic wounds.	[92]
PHBV/cellulose	Zinc Oxide nanocrystals	Antibacterial activity in wounds that are both acute and infected.	[93]
PLLA	Curcumin	Antioxidant, anti-inflammatory effects.	[94]
CA/polyester urethane	Polyhexamethylene biguanide	Antimicrobial activity.	[95]

**Table 5 pharmaceuticals-16-00325-t005:** Summary of various electrospun scaffolds in wound healing.

Electrospun Scaffolds	Main Findings	References
Electrospun fibers fabricated from a combination of Paclitaxel and (2-hydroxyethyl methacrylate)/bamboo cellulose were created.	This structure can be used to combat skin cancer and increase wound recovery.	[101]
Cellulose that has been carboxymethylated and CA electrospun fibers that are shaped like ribbons and contain silver nanoparticles.	The addition of silver nanoparticles improved the antibacterial and healing abilities of CMC fibers.	[102]
Nanofibrous networks fabricated of chitin and chitosan can be created by utilizing 1,1,1,3,3,3-hexafluoro-2-propanol as a spinning solvent.	Chitosan stimulates macrophages to aid in wound repair, causing polymorphonuclear neutrophils to move to the wound site at the initial time of healing.	[136]
PEO/chitosan/PCL/oliveoil (Composite fibers).	Composite fibers exhibited antibacterial action *S. aureus* and *E. coli*, a 0.6-fold decrease in edema, facilitated cell growth, and proliferation.	[106]
A fibrous mesh composed of both poly (glycerol sebacate) and poly L-lactic acid was created via coaxial electrospinning.	Exhibited cell proliferation, with a lower inflammatory response, re-establish pores and skin wound tissues.	[107]
Collagen-coated PHB/gelatin/ostholamide (OSA) Electrospun Nanofiber mesh	Nanofiber mesh shows excellent mechanical stability, reliable enzymatic breakdown, and effective antibacterial development towards *Pseudomonas aeruginosa* and *Staphylococcus aureus*.	[119]
Electrosprayed fiber networks composed of poly(3-hydroxybutyrate)/poly(3-hydroxyoctanoate-co-3-hydroxydecanoate) with a complex of chitin-lignin/glycyrrhizin acid.	CLA complexes useful bio-based mixtures for functionalizing pores and skin contact substrates in an in vitro skin model and helpful in wound healing.	[117]
PCL-gelatin electrospun nanoscaffolds incorporated with quercetin and ciprofoxacin hydrochloride (CH).	The full thickness wound was healed in 16 days.	[137]
Electrospun nanofibers containing Polyacrylic acid, and a synthetic biodegradable elastomer called poly(1,8-octanediol-co-citric acid).	Excellent antibacterial activity and delivery of physiologically relevant growth factor concentrations topically.	[132]
Electrospun mesh of polystyrene incorporated with p chamomile extract and poly(caprolactone).	Excellent wound-healing properties.	[129]

**Table 6 pharmaceuticals-16-00325-t006:** Various electrospun commercial wound dressings.

Product	Polymer	Device	References
Pathon	Polyurethane	Composite mesh, NO drug deliver	[140]
SurgiCLOT^®^	Dextran	Fibrin Sealant Patch	[141]
SpinCare™	Various electrospinnable polymers	Portable electrospinning wound dressing device	[142]
Tecophilic™	Polyurethane-PEG	Photosensitizing-loaded mesh	[143]

## Data Availability

Data sharing is not applicable.

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
