# Peer review of "Biodegradable Electrospun Scaffolds as an Emerging Tool for Skin Wound Regeneration: A Comprehensive Review"

_pharmaceuticals, 2023, doi:10.3390/ph16020325_

Round 1

Reviewer 1 Report

The review by Sharma et al. gives an overview on the advances in electrospun biodegradable polymers for wound healing application. Some major revisions should be performed before publication:

1. The authors are recommended to give the number of papers and patents (a graph could be added) in this field in the introduction in order to justify a review in this topic. The keywords used to make this graph should be added in the figure legend.

2. Some recent reviews in the area of electrospinning-based dressing for wound healing application like 10.1007/s11706-021-0540-1, 10.3390/nano12050784, 10.1007/s10965-021-02870-x should be mentioned in the introduction. A justification for an additional review in this field should be given.

3. A graphical abstract that can reflect the scope of the review well is recommended to be drawn and presented in this paper, which will be helpful for understanding of readers.

4. In the section 2, some more descriptions about the healing of chronic wounds like diabetic wounds should be added.

5. In the section 3, a schematic about the electrospinning process and device is recommended to be added.

6. The title of section 4 is too misleading, and it should be rewritten. What does “Biological Applications of Electrospinning Parameters” mean?

7. A table which summarize the application of bioactive molecules in electrospinning should be added in the section 5.2.

8. Some additional figures and tables are recommended in Section 6 to cover the whole area well.

9. Some statements feel they are lacking references. Admittedly, some of these statement might be considered well known facts, the concepts mentioned might have been referenced previously or will be in the future, but it might still be pertinent add references next to these statements for new readers to the field, skim readers or people who don't necessarily want to go looking for the relevant reference.

10. The authors should make a critical review instead of plain text flow. This means that a comparative discussion should take place assisted with categorization of the attributes of the different systems in each section/Tables.  

11. As well known that, nanomaterial-based strategies have been widely investigated for wound healing application. Do they have any commercial products based on nanomaterial technique available now?

12. The paper contains some typo and graphical errors. Please read carefully and correct Them.

Reviewer 2 Report

Dear Authors,

The purpose of the study entitled "Biodegradable polymers as an emerging tool for skin wound regeneration: a comprehensive review" is to review various wound healing materials based on biodegradable polymers.

In spite of the interesting issue, the Review in current form has major drawbacks, and I kindly recommend the Authors to revise the study due to the comments and suggestions.

1. Firstly, the title of the study does not summarize the main topic of the article: the Authors in general discuss electrospun nanofibers and nanofibrous materials based on polymers, while the Topic highlights the biodegradable polymers, which is confusing for the readers. Please, make necessary corrections.

2. Secondly, the structure of the study seems to be fragmented: some information about wound healing, some information about electrospinning, and examples of electrospun nanofibers based on different polymers without any logical explanation. The subsections 5.1. and 5.2. are not detailed and after the comprehensive analysis could be separate into individual reviews.

3. Usually, the reviews contain the comparative and detailed analysis in relation to the specific topic. Unfortunately, in this case the Authors do not follow this rule. There is a sense that the Authors only listed the results obtained without extensive analysis, although the Title of the manuscript promises the extended review. The Authors should answer several questions: how the biopolymers effect on the wound healing, how the microstructure (fiber diameter, porosity, permeability) effects on the wound healing process, how the additives loaded into nanofibers could influence on the wound. It is important to compare fibrous materials with film materials (if possible).

4. There are a lot of similar reviews in this topic, and the Authors tried to make an attempt to form a unique study. Unfortunately, the novelty of this review is not highlighted. Please see the links below. I hope that it helps the Authors to restructure the manuscript.

https://doi.org/10.3390%2Fjfb11030067

https://doi.org/10.1016/j.ijpharm.2019.05.053

https://doi.org/10.3390/membranes11090702

https://doi.org/10.3390/nano12050784

https://doi.org/10.1021/acsabm.8b00637

https://doi.org/10.3390/membranes11120908

5. The main text contains huge paragraphs, which should be divided into individual paragraphs. Moreover, I kindly recommend the Authors to include additional pictures, schemes, tables, etc. It is preferable to balance the text and figures in the good and illustrative Review.

6. Tables 1, 2, and 3 do not contain any references. At the same time, the necessity of Table 1 causes doubt for a couple of reasons: the influence of various parameters were discussed several times, and the list of parameters is not completed. Table 2 could be widened and changed into individual study (mini-review), which represents the modern electrospinning systems used for healing in situ.

7. Subsections 6.6-6.9 are too small and should be analysed in detail.

8. Reference list is very poor. In its current form, it looks like a mini-review. Please, add at least 20-30 related references, published in the last few years.

9. Conclusion must be widened. Moreover, this section in Reviews usually contains the future perspective, which demonstrates the possible ways of investigations in this area. Please, change the title of this section from “Conclusion” to “Conclusion and Future Perspectives” and demonstrate some ideas, statements, hypotheses, predictions etc. combined with the recommendations for topical elaboration and studies that are very important for our scientific colleagues working in this field.

10. The language of the article must be totally improved and corrected. In current form, the manuscript is hard-to-read. Please, try to use academic language and avoid informal language.

There are also several minor corrections, which should be taken into consideration:

1. Formatting of the article should be followed by the MDPI template. Please, carefully check and correct the subsections, tables, italic writing of in vivo and in vitro, as well as bacteria names, etc.

2. The main text contains several abbreviations and acronyms, which should be explained.

3. I kindly recommend the Authors to distinguish among concepts biodegradable polymers and electrospun nanofibers.

4. Figure 1 does not contain labels A-D, while the figure caption includes these ones.

5. P. 4, Table 1

Polymers are characterized by molecular weight, for this reason the term “atomic weight” is not correct.

6. P. 7, Figure 2

Why are the nanofibers based on chitosan-PVA-silk fibroin highlighted?

7. P. 7, Table 3

I kindly recommend the Authors to add some comparative analysis based on the study listed in this table. How do  therapeutics influence the electrospinning process and wound healing?

Based on the above-mentioned facts, unfortunately, I cannot recommend this study for publication in current form.

Reviewer 3 Report

The manuscript, "Biodegradable polymers as an emerging tool for skin wound regeneration: a comprehensive review," is not a comprehensive review of the subject. This paper provides an overview by summarizing some general data from published literature. The manuscript does not provide a more in-depth critical analysis of the advantages/disadvantages, potentials/limitations, opportunities/challenges of using biodegradable electrospun scaffolds for skin tissue regeneration and wound healing applications. Here are some suggestions for improving the quality of the manuscript:

1. The title should be changed to reflect the focus of this review paper on biodegradable electrospun scaffolds for wound healing applications. Electrospun scaffolds should be mentioned in the title.

2. The "Abstract" should include some background information on the electrospinning process and the biomedical applications of electrospun scaffolds.

3. Common therapeutic strategies for the treatment of skin wounds, the limitations of current approaches, the need for alternative approaches, the use of electrospun scaffolds for wound healing, and how these scaffolds overcome the limitations of current approaches should all be discussed in the "Introduction" section.

4. It is strongly suggested that the authors include a final paragraph in the "Introduction" section explaining the purpose of the study and the various sections of the paper, so readers know what subjects will be discussed in the rest of the paper.

5. The section "2. Healing of Wounds" is poorly written. The majority of the content is irrelevant to the topic. This section should highlight the events that occur during the four stages of the wound healing process. (For the reference: https://www.sciencedirect.com/science/article/abs/pii/S0141813021021541)

6. It would be helpful if the authors provided some background on skin structure before explaining the wound healing process.

7. The caption for Figure 1 should be improved. It would be helpful if the key events of each phase were mentioned.

8. It is suggested that the authors include a figure in the section "3. The Electrospinning Process and Its Benefits in Wound Healing Applications" that depicts various parts of an electrospinning apparatus as well as the electrospinning process.

9. Sections 3 and 4 could be combined.

10. It would be interesting if the authors included a separate section about desirable properties of scaffolds used for wound healing applications, emphasizing the role of electrospun scaffolds in this regard.

11. The environmental parameters should be included in Table 1.

12. Section 5 is poorly written. This section contains some general information on the delivery of antibacterial and bioactive agents during the wound healing process. A wide range of biodegradable scaffolds with various functions have been developed. There are many innovative drug delivery systems used for wound healing that the authors have not discussed, such as drug delivery using core-shell fibers, electrospun fibers for delivery of antibiotic-containing nanostructured lipid-based carriers (https://onlinelibrary.wiley.com/doi/full/10.1002/jbm.b.35039), bilayer or multilayer scaffolds incorporated with angiogenic or anti-inflammatory bioactive nanobiomaterials (https://www.mdpi.com/1999-4923/13/2/183), and so on.

13. References are required for the studies summarized in Table 3.

14. Section 6 only discusses a few common biomaterials used for skin tissue regeneration. The majority of the information in this section could be summarized in a single Table. This section also fails to provide a critical analysis of the advantages/disadvantages, potentials/limitations, and opportunities/challenges of the discussed electrospun scaffolds for wound treatment.

15. Many statements in the manuscript should be supported by relevant references.

16. Given the recent increase in the number of studies on the use of electrospun scaffolds for wound healing, the most recently published articles should be discussed.

17. The authors should seek editing assistance from someone with full professional proficiency in English.

Other minor revisions:

- "et al.," "in vitro," "in vivo," "Escherichia coli," "E. coli," "Staphylococcus aureus," and "S. aureus" should be written in italics throughout the manuscript.

- "Biodegradable polymers" is an appropriate keyword for this study.

- On page 2, "The human body is a multipurpose organ … synthetic, and natural impacts." Did the authors mean the skin when they mentioned "the human body"?

Round 2

Reviewer 1 Report

Some of reviewer's comments like questions 1, 2, 4, 5, 7, 11 are not well addressed, and the language should be polished again. Moreover, almost all the Figures are from other papers, do they get the copyrights? Some schemtics like the electrospinning setup are recommended to be redrawn by the authors.

Reviewer 2 Report

Dear Authors,

I am very grateful to you for the corrections of the manuscript.

The observations made on the study reviewed in an initial form were respected and remarkable improvements were made to the material of the manuscript.

At the same time I would like to suggest several additional changes.

1. Graphical abstract

It is preferable to use specific classification of the polymers: natural, semi-synthetic, and synthetic:

DOI: 10.1016/B978-0-12-813663-8.00002-6

DOI: 10.1088/978-1-6817-4079-9ch7

I kindly recommend the Authors to add “semi-synthetic” into the graphical abstract.

Also, please, add a comma after “Chitosan”.

2. P. 3, line 100

Please, add several examples of thermoresponsive polymers.

3. P. 6, line 249

The major quantity of solvent evaporates from the polymer solution before the fibers approach the collector. For this reason I recommend to change the phrase “As the solvent approaches the collector's surface, it evaporates” into “During the fibers approach the collector's surface, the solvent evaporates from them”.

4. P. 7, Figure 2

Please, correct “Tylor cone” to “Taylor cone” in the figure.

5. P. 7, line 276

Please, add the corresponding references after the cited authors: Deitzel et al. [Ref.], Meechaisue et al. [Ref.] and Zong et al. [Ref.]

6. P. 9, line 327

I suggest using the SI units of viscosity (Pas) instead of poise (cP).

7.  P. 9, line 330, subsection 4.5. Effect of solution conductivity

Please use the phrase “electrical conductivity” instead of “solution conductivity” or “conductivity of the solution” in this subsection.

In addition, I kindly recommend checking this parameter in Table 1 (P. 10-11).

8. P. 10, line 380

Please, add the reference after the author's “Park and Lee” [Ref].

Also, please, delete the year (2010) in line 382.

9. P. 13, Table 2

It would be very useful for readers if the Authors include in this Table additional information, for example, the solutions and the fiber diameters (diameter distributions). Moreover, the electrospinning parameters also could be included for the comparative purposes.

10.  Please, add the summarizing table related to the subsection 5.2, similar to the previous subsection.

11. I again kindly recommend the Authors add several images from the cited literature. The schemes, pictures, diagrams, etc. allow to increase the readability and the attractiveness of the manuscript.

12. P. 24, line 884

Staphylococcus aureus should be written in Italic.

13.  I kindly recommend the Authors to use the style format in accordance with the MDPI template. For example, References style must follow MDPI and ACS Style.

Reviewer 3 Report

The corrections and additions introduced by the authors improved the structure and quality of the manuscript. I have no further suggestions.

Author Response

Thank you for your valuable suggestions

Round 3

Reviewer 1 Report

The authors address most of the reviewer's comments. The quality of some figures like Figure 1, 4, and 5 is still needed to be improved. Moreover, the current challenges and future prospectives should be expanded and enhanced. In addition, the language is still needed to be polished.

Author Response

Dear Editor,

All the figures were made using Professional paid tools. We did not find any mistakes in Figure 1, 4 and 5. In fact, we believe that these figures are prompt in conveying our aim of the subject. But we are ready to correct the figures if the reviewer made any comments very specifically.

The current challenges and future prospectives are now improved in the revision file.

We have read the manuscript to remove any errors in the writing.

Reviewer 2 Report

Dear Authors,

Thank you very much for the corrections.

The manuscript was significantly improved and can be accepted in present form.

Author Response

Thank you for your comments